# Multi-Scale Mechanics of Cryopreserved Human Arterial Allografts Across a Six-Month Period

**DOI:** 10.3390/jfb16060198

**Published:** 2025-05-29

**Authors:** Gergely Imre Kovács, László Hidi, Evelin Forró, Dóra Haluszka, Dániel Sándor Veres, Gergő Péter Gyurok, Andrea Kőszegi, Attila Fintha, Miklós Kellermayer, Péter Sótonyi

**Affiliations:** 1Department of Vascular and Endovascular Surgery, Semmelweis University, 1085 Budapest, Hungary; kovacs.gergely.imre@semmelweis.hu (G.I.K.);; 2Department of Biophysics and Radiation Biology, Semmelweis University, 1085 Budapest, Hungary; 3Department of Heart Surgery, Semmelweis University, 1085 Budapest, Hungary; 4Department of Pathology and Experimental Cancer Research, Semmelweis University, 1085 Budapest, Hungary

**Keywords:** cryopreservation, vascular allograft, mechanics, nanoindentation, uniaxial ring test

## Abstract

Operating under septic conditions poses significant challenges in vascular surgery. Infection is a serious risk when handling synthetic vessel prostheses and is one of the most dreaded complications. In the event of graft infection, an infection-resistant alternative is necessary. Cryopreserved vascular allografts offer a suitable alternative to replace an infected vessel or a section of a synthetic graft. However, there are no international guidelines for the preparation, storage, and thawing of such vessel grafts. Here, we aimed to investigate the mechanical properties of human cryopreserved arteries across multiple scales, ranging from nanonewton to newton forces and identify the optimal cryogenic storage duration. Human arterial allograft samples were frozen in a slow, controlled process and stored at −80 °C. One native and four cryopreserved samples were examined during a six-month-long period. Dimethyl-sulphoxide was used as a cryoprotectant. The mechanical properties of fresh and stored samples were explored in uniaxial ring tests and nanoindentation. We found no significant changes in the multi-scale mechanical properties during the examination period. Our results indicate that the cryopreserved vascular allografts are mechanically stable for up to six months under cryogenic conditions; hence, they represent ideal samples in vascular surgery.

## 1. Introduction

In spite of careful procedures such as aseptic operations and antibiotic prophylaxis, infection remains a challenging complication in vascular surgery. In the commonly used surgical technique of prosthetic vascular graft implantation, infection may lead to a vascular graft infection with a ratio of up to 9% (0.5–9%) based on clinical reports [1,2,3]. In such situations, the implantation of vascular allografts (VAs) could be a viable alternative. Conservation and storage of VAs, however, poses technical challenges. The use of cryoconservation and modern cryoprotectants has facilitated the off-the-shelf availability of human VAs since the 1950s. Beyond improving accessibility, cryopreservation can reduce rejection reactions and other complications [4,5].

Its advantages notwithstanding, it is important to address the incidence of complications when applying cryopreserved vascular allografts (CPAs) under infectious conditions. The early (<30 days) mortality rate for CPA implantations ranges from 0.7% to 13%, and the occurrence of early graft-related complications, such as rupture and occlusion, reported in recent decades, is between 0% and 17% [1,6,7,8,9,10,11,12,13]. The incidence of complications depends on the type and condition of the CPA, as well as the anatomical location; for example, the complication rate after implantations may be higher in the case of peripheral operations [14,15]. As one of the worst and least coveted complications, early rupture or bleeding of CPAs have an incidence between 0.03 and 9.5% [8,9,10,11,12,16,17,18,19,20]. The reasons behind these complications include anastomosis disruption and spontaneous rupture of the graft material. These acute complications, together with chronic degeneration and the loss of the initial mechanical properties, may lead to aneurysm formation and cause critical complications such as delayed rupture [21,22]. To prevent the development of life-threatening complications, it is crucial to maintain the physiological properties and particularly the mechanical properties of the VA throughout the cryopreservation and storage process.

Although several procedures have been investigated previously, no international guideline is available to define the optimal preparation and proper circumstances of preservation for CPAs. Each cryopreservation method provides endless storage time in theory, but the published clinical results differ. Wang et al. compared the graft patency of cryopreserved allografts (CPAs) that had been stored for less than one year with those stored for more than one year, and found identical graft survival rates [23]. We note, however, that in this study the veins were stored for a shorter time and implanted sooner than the more durable arteries [23]. The overall frequency of graft-related complications increases after a graft spends more than six months in the cryopreserved state, based on our clinical experience spanning decades [9]. Previously, as part of research into the background of complications that increase with the length of storage, the thrombogenicity and platelet activation of CPAs were assessed, but no clinically significant differences were observed [24]. To identify the other possible causes of complications which are of mechanical nature, the investigation of the time-dependent mechanical properties of the arterial wall across multiple scales, from the nanoscopic to the macroscopic, across an extended storage period is required.

In line with this clinical observation, 6 months was chosen as the prospective cohort as this is a clinically relevant threshold at which a possible deterioration in mechanical properties can be detected. Furthermore, to extend our investigation beyond a simple comparison, several measurements were planned to determine any tendency in observed capabilities.

Exploring vessel specimens through mechanical testing is a well-established and reliable method for determining their elastic and tensile strength properties [25]. Tissue samples analysed using additional nanoscale methods, such as ultrasensitive nanoindentation tests, can provide more comprehensive results regarding the mechanical capabilities on a nanoscale level [26].

## 2. Aim

Our present research aimed at investigating whether the nanoscale to macroscale mechanical properties of VAs are affected by either freezing or storage time using our Cardiovascular Allograft Bank protocol.

## 3. Materials and Methods

In this prospective study, eleven femoral artery samples were harvested from multi-organ human donors and enrolled in this research. Samples from each arterial graft were examined five times during the six-month period following harvest and cryopreservation. Accordingly, five samples from each donor were analysed: (1) a native, fresh sample prior to cryopreservation (labelled “BC”); (2) a sample immediately following cryopreservation (“C0”); and samples stored for (3) one (“C1”), (4) twelve (“C12”), and (5) twenty-four (“C24”) weeks.

### 3.1. Donor Inclusion and Exclusion Criteria

Donor inclusion criteria were based on the national multi-organ donation criteria and rules [27]. During explantation, all VAs were investigated for approval by a vascular surgeon specialist (e.g., significantly injured or calcified VAs were excluded). All inclusion-exclusion criteria have been published previously [24].

### 3.2. Ethics Approval

Our study fully complies with the principles of national multi-organ donation and the applicable international and national laws and regulations. The anonymized data of the donors were collected prospectively from the electronic health information system of the donation according to the General Data Protection Regulations of the European Union. The institutional review board of Semmelweis University Regional and Institutional Committee of Science and Research Ethics approved the study protocol (approval number: 257/2018). Patient consent was not obtained as the vascular tissues were harvested from brain-dead, multi-organ donors, and the data were analysed while maintaining anonymity throughout the investigation [24].

### 3.3. Cardiovascular Allograft Biobank Protocol: Harvest, Cryopreservation, Storage, and Thawing of CPA Samples

Femoral allografts were explanted from brain-dead donors under sterile conditions, and placed in 500 mL of cooled, saline-based (Sodium Chloride 0.9% “Baxter” Intravenous Infusion in Viaflo, Baxter Hungary, Budapest, Hungary) transport solution containing 0.4 mg/mL fluconazole (Fresenius Kabi Hungary, Budapest, Hungary) and 4 mg/mL cefazolin (Sandoz GmbH, Kundl, Austria) at 4 °C in the laboratory for the subsequent experimental steps. Cryopreservation was performed in a sterile environment within 12 h. After secondary quality checks and microbial testing, the fresh artery samples were placed in a saline solution and were transported immediately to the testing sites at 4 °C temperature. Mechanical tests were conducted within 12 h of sample collection (BC). Artery samples, except for the native specimen (BC), underwent controlled cryopreservation in a cryoprotectant solution containing 20% (*v*/*v*%) dimethyl sulfoxide, 4 mg/mL cefazolin, and 0.4 mg/mL fluconazole. The temperature steps of the controlled freezing process are shown in Figure 1. Four CP samples were produced from one donor and thawed after the following times during the investigation period: (1) immediately following freezing (C0) and (2) one (C1), (3) twelve (C12), and (4) twenty-four weeks (C24) after cryopreservation. Thawing was carried out by warming the samples in a 37 °C water bath. To remove residues of the cryoprotective agent, the samples were rinsed three times with saline solution. Further details of the applied cardiovascular allograft protocol were reported previously [24].

### 3.4. Uniaxial Ring Test

The uniaxial ring tests were conducted in physiological saline at room temperature and atmospheric pressure using an Instron 5942 mechanical tester with a 50 N load cell, which allows unidirectional, strain-controlled extension (Figure 2). The load cell accuracy was equal to or better than 0.25% of the indicated force.

Two hooks gripped the 5 mm-wide rings, as shown in Figure 2b. The tensile strength and rigidity tests were carried out in one measurement, which consisted of successive cycles of extension and relaxation with progressively increasing length. Once the target extension was reached, the sample was relaxed by returning the probe to its starting position. These turning points were at 2, 4, 6, and 8 mm. Before starting each new cycle, the specimen rested in its initial position for 5 s. Finally, in the last mechanical cycle, the sample was stretched until it ruptured, thus providing information about tensile strength. The stretch and relaxation rates were 1 mm/min throughout the tests.

To characterise the mechanical properties of the arterial wall, the following five parameters were calculated from the force-displacement curves [25]:Tensile strength ($F_{TS}$);Stretch ratio at maximum load ($\lambda_{f}$);Extensibility;Engineering stress (P);True stress (σ).

Tensile strength ($F_{TS}$) is the ratio of maximum load at wall rupture ($f_{max}$) and the volume of the arterial wall ($V_{art}$), and is expressed as follows:(1)$$F_{TS}=\frac{f_{max}}{V_{art}},$$
where *V*_art_ is calculated from the geometric parameters of the arterial ring using the following equation:(2)$$V_{art}=2\left(r\pi xl+h_{0}xl\right),$$
where the *r* is the radius of the wire, and *x*, *l*, and *h*_0_ are the specimen’s width, thickness, and initial length, respectively (Figure 3).

The stretch ratio ($\lambda_{f}$) in the circumferential direction was approximated by the following equation:(3)$$\lambda_{f}=\frac{h_{f}+\pi r}{h_{0}+\pi r},$$
where $h_{f}$ and $h_{0}$ are the lengths of the arterial ring at *f*_max_ and in the initial position, respectively, and r is the radius of the wires (Figure 3).

To determine the extensibility of the arterial wall, Equation (3) was used as well, but in this case, $h_{f}$ was replaced by the maximum length at which the mechanical cycle was still reversible.

The engineering or circumferential stress (P) was calculated using the arterial cross-section which resists the pulling force, as show in the following equation:(4)$$P=\frac{f_{max}}{2xl},$$
where $f_{max}$ is the maximum applied load (see Equation (1)) and x and l are the specimen’s width and thickness, respectively (Figure 3). True stress (σ) was calculated using the exact extensibility according to the stretch ratio as follows:(5)$$\sigma =\frac{f_{max\;}\lambda_{f}}{2xl}.$$

### 3.5. Atomic Force Microscopy (AFM): Topography and Nanoindentation

In AFM experiments we scanned the sample surface with a cantilever with a tip to obtain topological features, which were mapped with nanometre resolution [28]. AFM imaging and indentation measurements were performed with an Asylum Research MFP3D instrument (Asylum Research, Santa Barbara, CA, USA). Non-contact scanning was carried out in air with an AC160 (k = 26 N/m) cantilever (Oxford Instruments, Santa Barbara, CA, USA). First, by using transmitted-light illumination, the tunica adventitia was identified, then the cantilever was positioned onto this region and 20 × 20 µm images were recorded (Figure 4). Following the determination of surface characteristics, indentations were performed in the same area in a 32 × 32 matrix, yielding ~1024 force curves per sample. For nanomechanical measurements, the surface-fixed sections were manipulated by first pressing the cantilever tip against the sample to obtain indentation force response, then pulling the cantilever away with a constant, pre-adjusted rate to obtain elastic stretch force curves. In each force map, a maximal indentation force of 500 nN was applied. From the force curves collected in each pixel, the Young’s modulus (YM) and stiffness were determined by fitting with the Hertz model. Image post-processing and data analysis were performed using the AFM driving software AR16, IgorPro 6.37 (Wavemetrics, Lake Oswego, OR, USA).

#### Sample Preparation for Nanoindentation Tests

Samples were embedded in Shandon cryomatrix gel (Thermo Scientific, Waltham, MA, USA) and incubated for 15 min, then sections of 20 µm thickness were sliced with a cryomicrotome and deposited onto poly-L-lysine coated slides, which were finally fixed in acetone for 10 min. Prior to AFM measurements, the samples were rehydrated in PBS for 5 min.

### 3.6. Statistical Analysis

As our study focused on differential comparisons, we employed a Bland–Altman plot (BAP) to analyse data. The results of cryopreserved (CP) samples thawed at different times were compared with the native samples. A BAP, or difference plot, is a graphical technique that compares two statistical sets of measurements. The differences between the two measurements are plotted against their means.

Horizontal lines are drawn at the mean difference and the limits of agreement (red dotted line in our plots), which are determined as the mean difference (blue line in our plots) ± 1.96 times the standard deviation of the differences [29]. Agreement between the measurements was established based on the following criteria:If there is no observable trend or pattern in the differences as the means increase.If the 95% confidence interval of the mean differences includes the value of zero or the difference is smaller than the measurement variability and precision, or if it is not clinically significant.If the 95% limits of agreement are within the range of the measurement variability and precision, or if the difference is not clinically significant.

Further to utilising the BAP, the mean, the distribution, and the standard deviation were calculated for each force map of the tunica adventitia in the case of AFM measurements and for the uniaxial ring tests. These values were then plotted in diagrams for each storage period (C0–C24).

Data analysis was conducted using Microsoft Excel (v16.97), and BAPs were generated using R software (v4.4.1). Any other figures were edited in Microsoft PowerPoint (v16.97).

## 4. Results

### 4.1. Donor Characteristics

The median age of the donors was 45.0 (33.0–50.5) years. The leading cause of death was cerebral haemorrhage (6 donors; 54.54%). Three donors were female (27.27%). The median body mass index (BMI) was 26.3 (23.65–27.8) kg/m^2^. Table 1 summarises the most important and clinically relevant features of the donors.

#### 4.1.1. Tensile Strength

According to the BAP, there was no significant deviation in the mean tensile strength of the preserved arteries compared to the native vessel. At the end of the first week of cryopreservation, the difference between the tensile strength of the samples decreased slightly (mean difference = −0.0580 ± 0.0610), but the deviation is clinically irrelevant. During further storage, the mean difference in tensile strength values increased to the initial level. Tensile strength did not change considerably until the end of the observation when compared to the native sample (Figure 5). The agreement limit has essentially an identical range at each measured series (Figure 5), so the scale of dispersion is similar at each comparison.

#### 4.1.2. Stretch Ratio at Maximum Load

The tension of the examined arteries did not change significantly either when compared to the native vessels. The mean tension difference fluctuated around the baseline (Figure 6) and showed no trend (Table 2). The limits of agreement were equal on each plot.

#### 4.1.3. Extensibility

The preserved vessels kept their extensibility, hence flexibility: the mean extensibility did not show significant change. Although the mean did not show considerable fluctuation, the agreement limit in one case (twelve-week sample) had a wider range (Figure 7 and Table 2).

#### 4.1.4. Engineering Stress (Circumferential Stress)

When comparing the stress load on the cross-sectional area of vessel walls, we found no significant difference relative to the native sample (Figure 8).

#### 4.1.5. True Stress

The true stress values were identical to the tensile strength results: we observed a slight increment on the first week of storage (mean difference = −1.2918 ± 1.4919). This difference was remarkable, and similarly to the tensile strength results after the 12th week, it provided a value comparable to that of the native sample (Figure 9).

Based on the ring-test measurements, no significant differences were found in the individually determined parameters. The samples kept their extensibility and load capacity along the cryopreservation. The difference values shown in the BAPs are summarised in Table 2.

### 4.2. AFM Nanoindentation

To determine the nanomechanical consequences of various storage times, we carried out nanoindentation experiments with AFM. First, a 20 µm × 20 µm area of tunica adventitia was scanned to obtain information about the topographical structure of the surface. Then, 500 nN maximal indentation force was applied on the same region of interest to collect force curves in each pixel in a 32-by-32 matrix. We obtained YM maps and corresponding distributions by fitting the Hertz model onto the nanoindentation force curves. Based on the nanoindentation test results, each section showed a broad distribution of YM with a high standard deviation without remarkable differences between the compared storage times (Figure 10 and Figure 11).

This tendency of data was confirmed by BAP as well. Based on the plot, no significant trend or correlation was found regarding the magnitude of YM between storage times. Mathematical functions could not be fitted to these points. The average differences decreased, meaning the overall YM value increased by the time compared to the initial (C0) time value. However, the standard deviations were also elevated, resulting in non-significant differences between storage time values. These results may lead to unambiguous statistical estimation. The difference values showed on Figure 12 are presented in Table 3. Raw results can be found in Appendix A.

## 5. Discussion

During the detailed mechanical characterisation of CP human arterial allografts, we did not find any significant or clinically relevant change regarding the ultimate strength, rigidity, or nanomechanical capabilities of the examined vascular samples under the investigation period.

Prior investigations of the mechanical properties of CP cardiovascular allografts often focused on comparing different freezing, preserving, or thawing methods to determine the quality of allografts [30,31]. According to previous reports, the main reason for injury during cryopreservation is the formation of extracellular ice crystals that cause a direct disruption of tissue integrity [32]. Alongside freezing, rapid thawing between −196 °C and −100 °C can also cause significant structural changes in human CPAs [33]. The rapidly introduced large amount of energy can evoke tension in the frozen material, damaging both the elastic and collagen fibre structures in the extracellular matrix (ECM). To reduce these effects, our protocol applied −80 °C preservation temperature using vaporised nitrogen following controlled freezing steps, and the thawing process was implemented gently.

Prior investigations on the possible impact of storage time on allograft quality and shelf-life are so far limited. For this very reason, a comprehensive series of clinical trials was launched to investigate the potential background for the observed time-dependent phenomenon, as there are relatively few studies in the international literature on the effect of storage time and no previous experimental studies on healthy human samples.

While several studies have explored cryopreservation in animal models or have focused on cellular and histological characteristics, to our knowledge, no prior research has specifically examined the mechanical integrity of human vascular allografts across multiple time points during storage.

Pukacki et al. examined the influence of cryopreservation on human iliofemoral arterial sections and published identical results comparing elastic moduli and compliance values in native and CP samples [34]. Placing load in the circumferential orientation of the vascular wall showed no difference in the elastic moduli and compliance of CPA compared to native samples, suggesting that the exact impact of cryopreservation on the elastic capabilities of arterial allografts is identical to our current findings. Unfortunately, neither temporal observations nor the tensile strength measurements of the examined graft sections were reported, although these can reveal crucial properties concerning the development of mechanical complications with implanted CPAs.

The mechanical examination carried out by Novotny et al. did not reveal any differences in ultimate tensile strength between 37 °C- or 5 °C-thawed human arterial allografts [35]. However, scanning electron microscopy revealed minor structural changes when comparing two different thawing protocols, although the bi-axial mechanical testing did not reveal a significant change in tensile strength. Besides the thawing rate that influenced the microstructural construction of arterial walls, the rapid freezing rate can also break the structural integrity of arterial grafts without changing their mechanical capabilities [36].

O’Leary et al. reported promising results on the long-term viability of VA in 2014 [37]. They placed porcine aortic tissue in saline solution and froze the samples to −20 °C. During a year-long observation, they performed bi-axial mechanical testing at five different times and found that the elastic moduli of VGs did not change over time [37]. Although this temporal observation would be beneficial from a clinical point of view, freezing in saline solution significantly increases the incidence of complications [38]. Accordingly, due to poor outcomes after implanting a VA preserved frozen in saline solution, it was not widely studied in clinical use.

Whereas multiaxial measurements can provide more information about the mechanical capabilities of samples, uniaxial tests using both ring and flat-strip specimens also can be a proper method for comparative examination, especially when a tissue sample is too scarce or small to conduct other types of mechanical testing [25]. The previously mentioned findings suggest that determining mechanical functions using only tensile testing is insufficient. Examining samples using a higher-resolution technique, which may validate the SEM findings and provide more detailed information about the nanomechanical capabilities of the examined tissues, can be more beneficial.

In addition to mechanical testing, determining the nanomechanical capabilities of a CPA can help to thoroughly characterise the impact of CP at the macromolecular level. It has been observed that the lack of cryoprotectant can increase YM over time during cold storage. However, relaxation and elastic modulus remained the same when the cryoprotectant was administrated [39]. According to our results, we did not find any significant alteration (especially right after the freezing), but a slight increase in the nanoscale stiffness of the adventitia was detected during the observed period.

The findings most relevant to our results were presented by Lőrinczy et al. They found that the cryopreservation time can modify the molecular stability of preserved porcine aortic grafts. They found no major alteration except for an initial decrement in thermal stability that returned to its initial value by the third month of preservation [40].

Our data show a slight increment in the tensile strength at the initial period of the observed interval. By the 12th week, it returns to its initial value and remains unchanged until the end of the 24-week follow-up. Lőrinczy’s research group extended the follow-up period in their subsequent study, confirmed the same results as before, and published another significant observation: a systematic decrease occurs in the thermal stability in the analysed arterial grafts after the 12th week of storage. According to our findings, there may be a relationship between the decrease in thermal stability and the nanomechanical results. However, our results do not indicate a significant decrease in the elastic modulus of the arterial allograft wall, but the average modulus showed an increasing trend, suggesting an increasing rigidity [41].

The results of the nanomechanical testing and the enhanced thermal instability over time indicate changes in the ECM. However, when evaluating the mechanical tests, we must carefully assess their clinical relevance. While our study provides insight into the mechanical stability of cryopreserved vascular allografts, it is important to emphasise that mechanical assessment alone does not permit definitive conclusions regarding the functional integration of CPAs following implantation. Graft performance also depends on cellular viability and host–graft interactions. Several studies have demonstrated that cryopreservation can lead to considerable cellular impairment, particularly affecting cellular viability. Cai et al. reported that prolonged cryopreservation of human umbilical vein endothelial cells resulted in compromised angiogenic potential, diminished excretory functions, but increased ICAM-1 expression [42]. Wang et al. found that smooth muscle cells retained their regenerative potential more effectively when using 1.5 M 1,2-propanediol as CPA, compared to DMSO application, which significantly impaired regeneration. Furthermore, mechanical properties were assessed and showed a significant decrease after cryopreservation with 1,2-propanediol [43]. Unfortunately, no mechanical tests were conducted with samples cryopreserved with DMSO. In light of the above results, it is not entirely clear whether there is a correlation between mechanical function and cell viability, so this question must be investigated further. While our current results contribute valuable baseline data on mechanical performance, future investigations, including cell viability and histological analyses before implantation and after in vivo graft failure, would be crucial for a more comprehensive understanding of graft function, long-term outcomes, and in vivo effects.

To reach further conclusions regarding the self-life of human CPAs, a more extensive prospective investigation is required. While our current study was not designed to directly assess in vivo outcomes, the preservation of mechanical properties during the 6-month storage period suggests that these grafts retain sufficient structural integrity to resist deformation and withstand hemodynamic forces following implantation. This preserved stability may reduce the risk of complications such as aneurysmal degeneration or anastomotic failure. Future studies integrating biomechanical data with clinical follow-up will be essential to confirm these implications. The mechanical properties of the implanted grafts could be determined several times during storage and even before implantation. This would allow us to compare the complication rate of these grafts with the mechanical results and the length of storage.

The main strength of the current study is that healthy human samples were analysed from one donor at consecutive time points. There are limitations, however. The analysed vascular rings were not completely identical. Because of the destructive measurements, we used consecutive sections of femoral arteries, which may introduce variability in our results. Furthermore, the number of samples was limited by the number of multi-organ donations. This serious difficulty caused an issue in one case: Unfortunately, we could not avoid losing one fresh sample from AFM measurements. The fresh sample was too disintegrated by the time of measurement, and according to its nature, no spare sample was available. The sample count was inherently limited by the low number of brain-dead organ donations, and additional ethical and clinical constraints further restricted sample collection. Since these allografts were designated for clinical use in daily practice, we were required to excise the smallest arterial segments possible for research purposes. To preserve clinical utility, the sampling was limited and reasonably balanced between scientific thoroughness and clinical feasibility. Finally, even larger time-scale measurements are required to assess the potential time-dependent degradation of VA quality. The present findings provide a solid foundation for these long-term studies.

## 6. Conclusions

According to the results of our multi-scale mechanical tests, the cryopreservation procedure applied here did not significantly modify the mechanical properties of the preserved grafts during the 24-week-long observation period.

The nanoindentation tests revealed a slight increase in elastic modulus overall, but its clinical relevance, evaluated together with the uniaxial ring tests, appears negligible. Altogether, the time factor may exert less influence on the mechanical properties than the various freezing and thawing methods.

Understanding the long-term effects of cryopreservation on vascular tissue integrity remains an area to be explored from other aspects to optimise allograft production and extend storage time to further improve their clinical accessibility and durability.

## Figures and Tables

**Figure 1 jfb-16-00198-f001:**
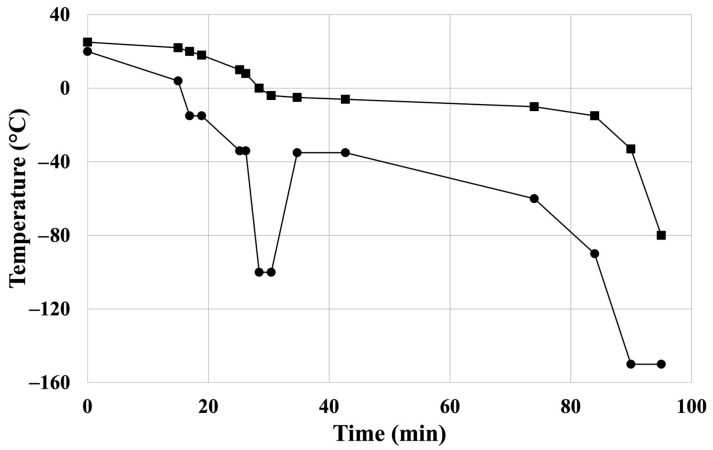
Controlled freezing protocol of the cryopreservation: temperature ramps of controlled-rate freezer’s chamber: “●”; temperature ramps of sample: “■”. Reprinted from Ref. [24].

**Figure 2 jfb-16-00198-f002:**
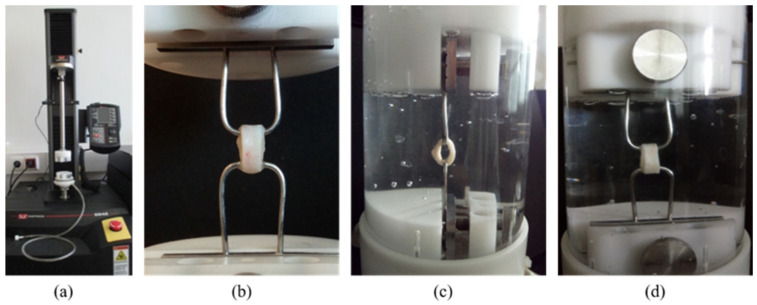
The Instron 5942 equipment (**a**), the ring sample gripped by 2 hooks (**b**), and the sample during the test in physiological saline; side view (**c**) and in front view (**d**).

**Figure 3 jfb-16-00198-f003:**
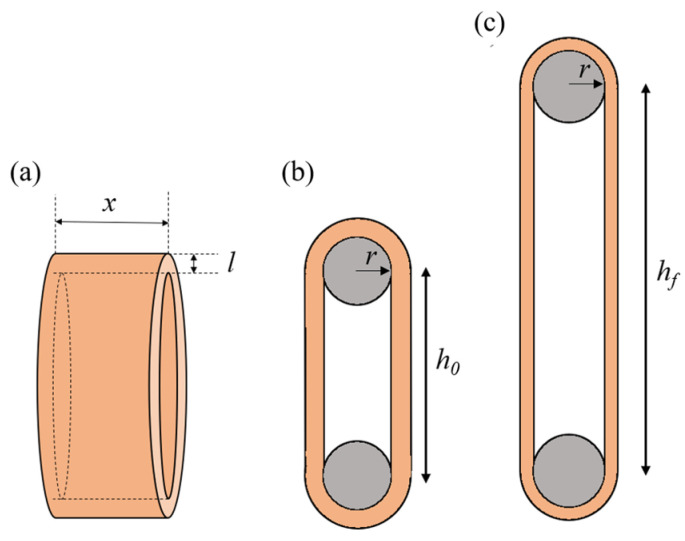
Schematics of the arterial ring (**a**). Illustration of the uniaxial tension test on an arterial ring, showing the initial configuration (**b**) and the deformed configuration (**c**).

**Figure 4 jfb-16-00198-f004:**
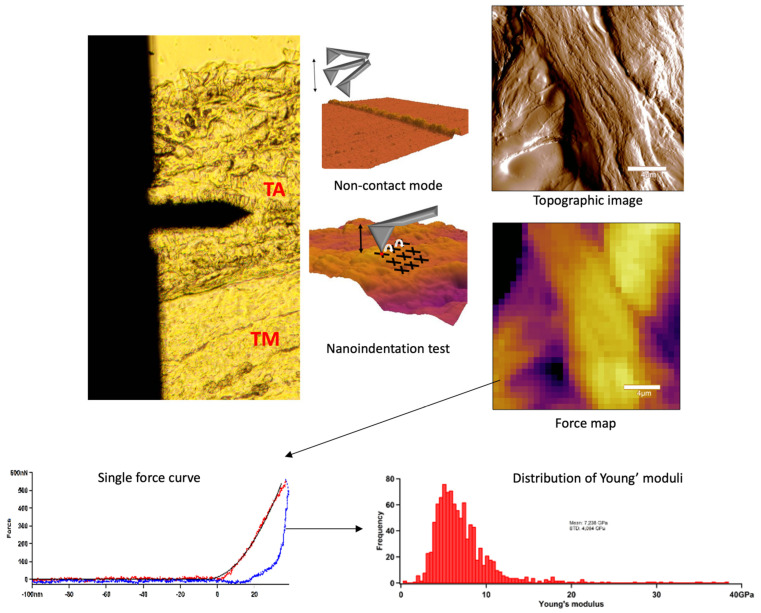
Summarised schematic figure about AFM measurements. First, tunica adventitia was detected, and then the surface was scanned with AFM non-contact mode. Following the topographical imaging, the nanoindentation test was applied. Force curves were acquired and the distribution of Young’s modulus was determined.

**Figure 5 jfb-16-00198-f005:**
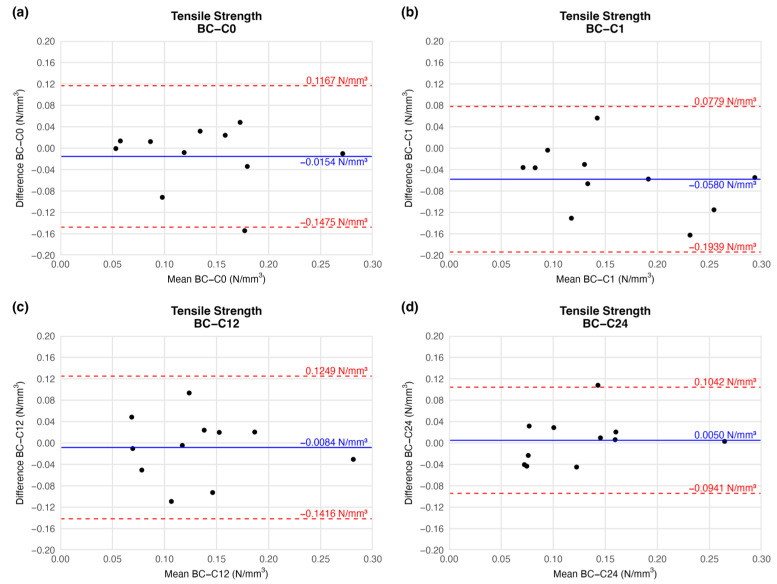
Bland–Altman plots of the tensile strength of the preserved arteries compared to the native vessel: native–after the freezing procedure (**a**), native–1 week (**b**), native–12 weeks (**c**), and native–24 weeks (**d**).

**Figure 6 jfb-16-00198-f006:**
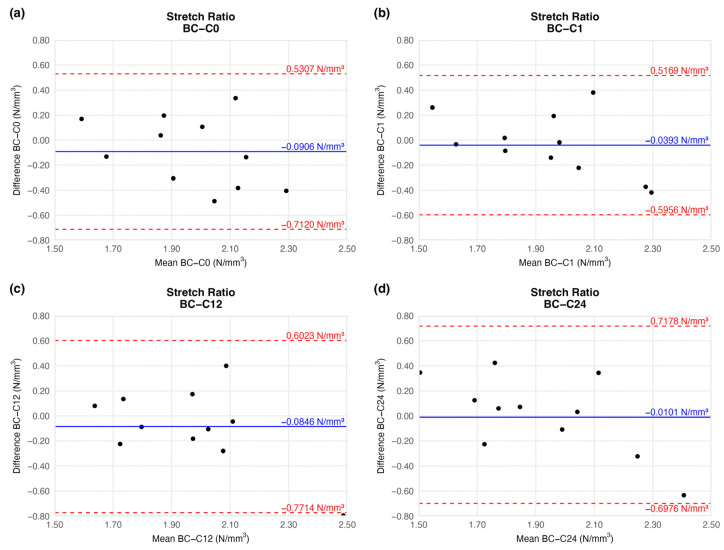
Bland–Altman plots of the stretch ratio of the preserved arteries compared to the native vessel: native–after the freezing procedure (**a**), native–1 week (**b**), native–12 weeks (**c**), and native–24 weeks (**d**).

**Figure 7 jfb-16-00198-f007:**
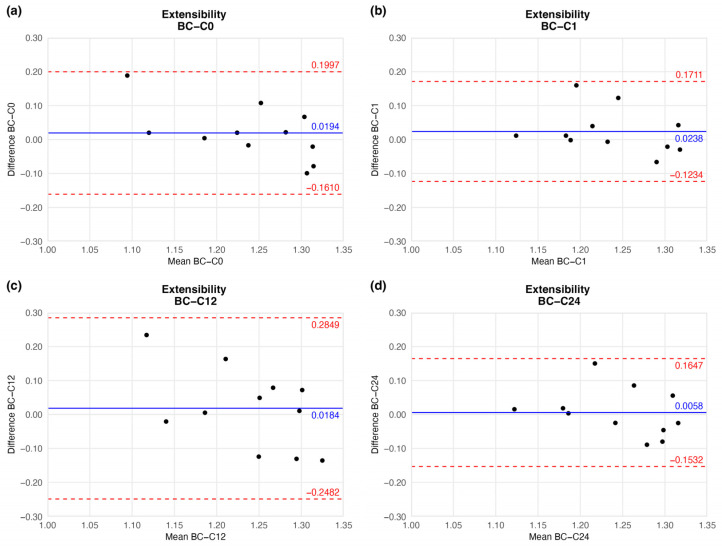
Bland–Altman plots of the extensibility of the preserved arteries compared to the native vessel: native–after the freezing procedure (**a**), native–1 week (**b**), native–12 weeks (**c**), and native–24 weeks (**d**).

**Figure 8 jfb-16-00198-f008:**
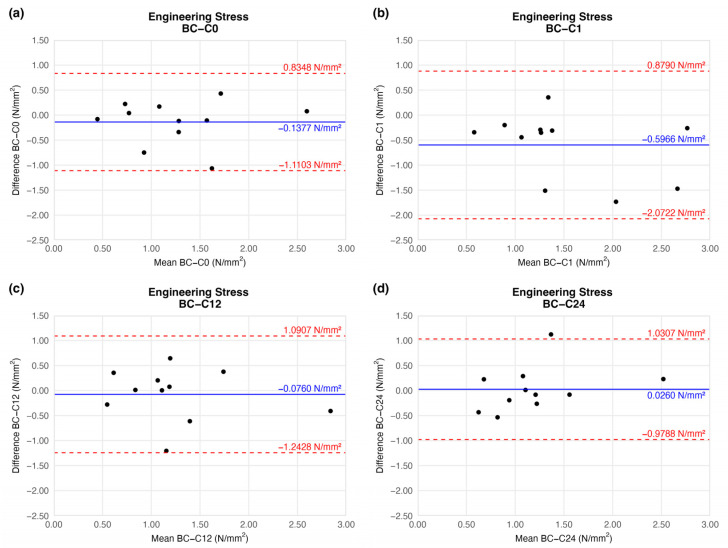
Bland–Altman plots of the engineering stress of the preserved arteries compared to the native vessel: native–after the freezing procedure (**a**), native–1 week (**b**), native–12 weeks (**c**), and native–24 weeks (**d**).

**Figure 9 jfb-16-00198-f009:**
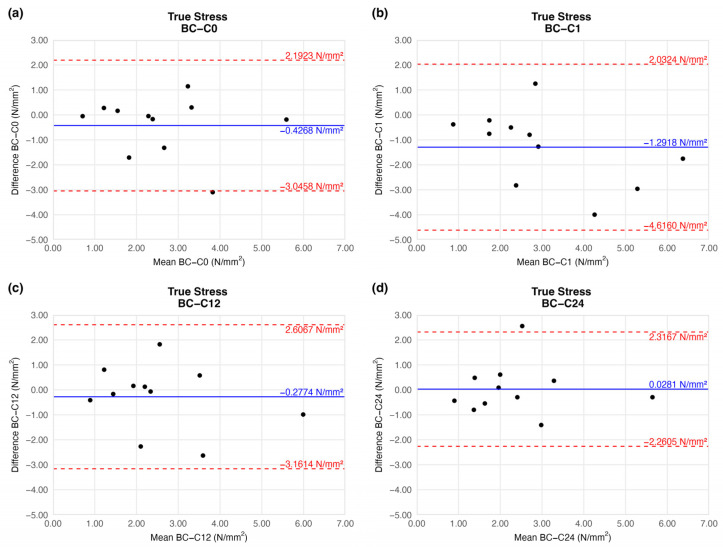
Bland–Altman plots of the true stress of the preserved arteries compared to the native vessel: native–after the freezing procedure (**a**), native–1 week (**b**), native–12 weeks (**c**), and native–24 weeks (**d**).

**Figure 10 jfb-16-00198-f010:**
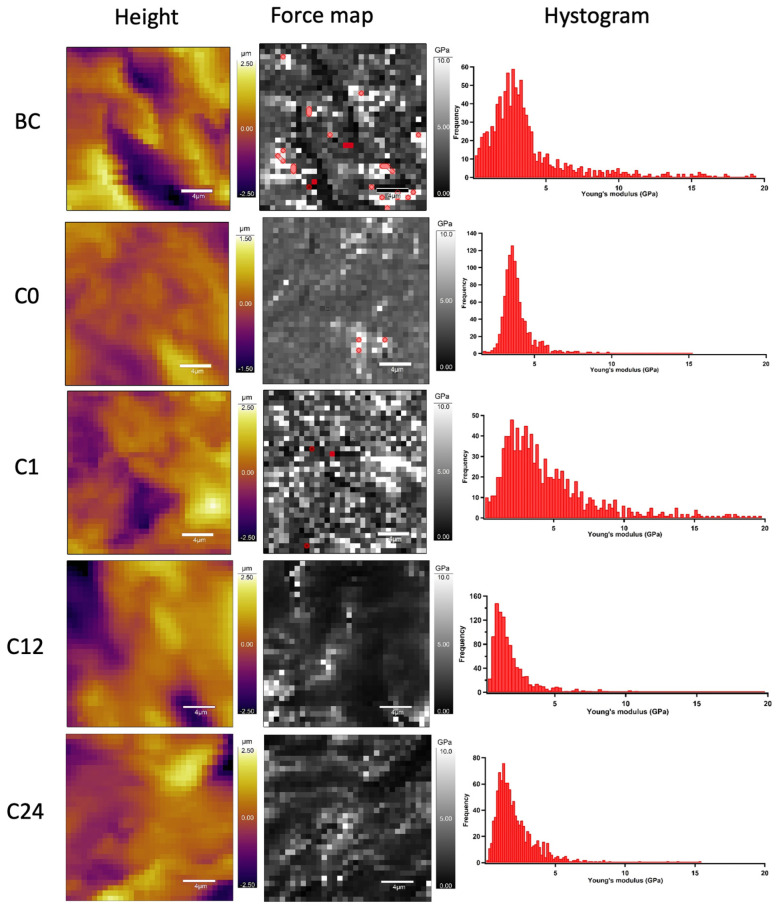
Representative image of AFM nanoindentation analysis in each time value. Height images show the topographical structure of surface in each pixel of 32 × 32 matrix. Converted Young’s modulus values in same pixels (force map). Red pixels were excluded from the analysis due to the wrong Hertz model fit (false force curves). Histograms show distributions of Young’s modulus on TA samples.

**Figure 11 jfb-16-00198-f011:**
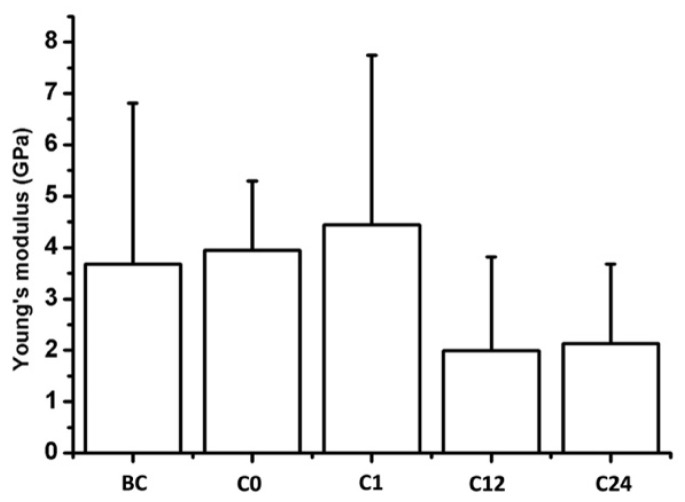
Mean Young’s modulus of the samples. Bars represent the average and error bars show the standard deviation (SD).

**Figure 12 jfb-16-00198-f012:**
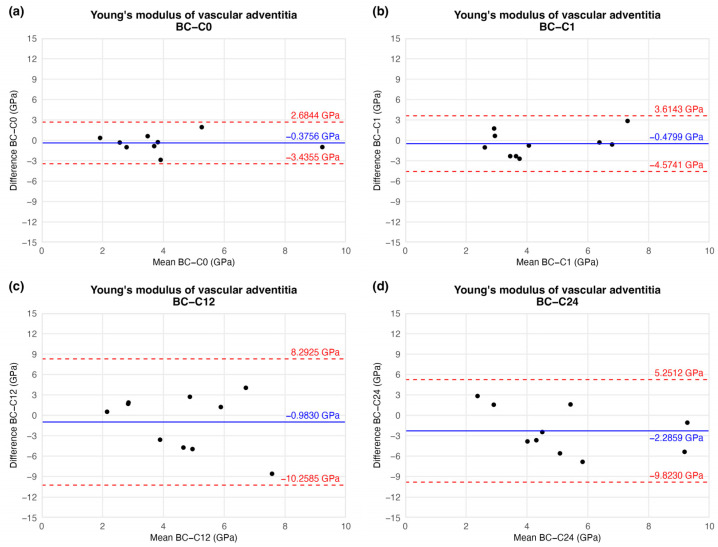
Results of Bland–Altman plot. Black points show the difference between two Young’s modulus values of two-time values as a function of the mean Young’s modulus of two measurements: native–after the freezing procedure (**a**), native–1 week (**b**), native–12 weeks (**c**), and native–24 weeks (**d**).

**Table 1 jfb-16-00198-t001:** Donor characteristics.

Characteristic		*n* = 11
Cause of death	Trauma	3 (27.27%)
	Cerebral ischaemia	2 (18.18%)
	Cerebral haemorrhage	6 (54.55%)
Age (years)		45.00 (33.00–50.50)
Female sex		3 (27.27%)
Body mass index (BMI kg/m^2^)		26.30 (23.65–27.80)
Medical history	Hypertension	3 (27.27%)
	Diabetes	1 (9.09%)
	Pulmonary disease (COPD)	1 (9.09%)
	Smoking	4 (36.36%)
Blood group	A	5 (45.46%)
	B	3 (27.27%)
	AB	0 (0%)
	0	3 (27.27%)
	Rh+	10 (90.91%)
	Rh−	1 (9.09%)

**Table 2 jfb-16-00198-t002:** Results of biomechanical tests. The table contains the difference between the results of the native sample and cryopreserved specimens.

	Elapsed Weeks	MD ± SD	CI 95% LL	CI 95% UL
Tensile strength (N/mm^3^)	C0	−0.0154 ± 0.0593	−0.1475	0.1167
	C1	−0.0580 ± 0.0610	−0.1939	0.0779
	C12	−0.0084 ± 0.0598	−0.1416	0.1249
	C24	0.0050 ± 0.0445	−0.0941	0.1042
Stretch ratio at maximum load	C0	−0.0906 ± 0.2789	−0.7120	0.5307
	C1	−0.0393 ± 0.2496	−0.5956	0.5169
	C12	−0.0846 ± 0.3083	−0.7714	0.6023
	C24	0.0101 ± 0.3176	−0.6976	0.7178
Extensibility	C0	0.0194 ± 0.0809	−0.1610	0.1997
	C1	0.0238 ± 0.0661	−0.1234	0.1711
	C12	0.0184 ± 0.1196	−0.2482	0.2849
	C24	0.0058 ± 0.0713	−0.1532	0.1647
Engineering stress (N/mm^2^)	C0	−0.1377 ± 0.4365	−1.1103	0.8348
	C1	−0.5966 ± 0.6623	−2.0722	0.8790
	C12	−0.0760 ± 0.5236	−1.2428	1.0907
	C24	0.0260 ± 0.4509	−0.9788	1.0307
True stress (N/mm^2^)	C0	−0.4268 ± 1.1754	−3.0458	2.1923
	C1	−1.2918 ± 1.4919	−4.6160	2.0324
	C12	−0.2774 ± 1.2944	−3.1614	2.6067
	C24	0.0281 ± 1.0271	−2.2605	2.3167

**Table 3 jfb-16-00198-t003:** Results of nanoindentation tests. The table contains the difference between the results of the native sample and CP specimens.

	Elapsed Weeks	MD ± SD	CI 95% LL	CI 95% UL
Young’s modulus (GPa)	C0	−0.3756 ± 1.3269	−3.4355	2.6844
	C1	−0.4799 ± 1.8099	−4.5742	3.6143
	C12	−0.9830 ± 4.1629	−10.2585	8.2925
	C24	−2.2859 ± 3.3827	−9.8230	5.2512

## Data Availability

The original contributions presented in this study are included in the article and Appendix A. Further inquiries can be directed to the corresponding author.

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
