# Peer review of "Multi-Scale Mechanics of Cryopreserved Human Arterial Allografts Across a Six-Month Period"

_jfb, 2025, doi:10.3390/jfb16060198_

Round 1

Reviewer 1 Report

Comments and Suggestions for Authors

Although this manuscript is well-written and presents valuable insights, there are a few issues that need to be addressed for further clarity. First, the rationale behind selecting a 6-month duration for the cryopreservation needs to be clearly explained. It would be helpful to provide a justification for why this specific timeframe was chosen, as the duration of cryopreservation can significantly impact the results. Additionally, the sample size is relatively small, which may limit the ability to draw meaningful or statistically significant conclusions. To strengthen the study, it is recommended that data from a larger number of samples be included to enhance the reliability and generalizability of the findings. Furthermore, the manuscript does not provide detailed information on how cell viability was assessed after cryopreservation. Given that viability is a critical factor in determining the success of cryopreservation, it is essential to clarify the methods used to evaluate this parameter. This additional information will help improve the overall understanding of the cryopreservation process and its effectiveness

Author Response

Please see the attachment for detailed answers and the updated manuscript with tracked changes.

Reviewer 2 Report

Comments and Suggestions for Authors Dear Authors, Thank you for the opportunity to review your manuscript entitled: “Multi-scale mechanics of cryopreserved human arterial allografts across a six-month period” Your study addresses a clinically relevant and timely topic in the field of vascular surgery and tissue engineering. The need to better understand the mechanical stability of cryopreserved vascular allografts over time is well justified, especially given the absence of standardized international protocols for their preservation and clinical use. The manuscript is well written, and the methodology is robust. The selected mechanical tests provide a comprehensive analysis that reinforces the reliability of your conclusions. Nevertheless, I would like to highlight several aspects that, in my opinion, could improve the overall impact and clarity of the work: Context and novelty Although the study is well designed, there are previous investigations that have addressed the mechanical effects of cryopreservation on vascular tissues. It would be advisable to more clearly specify in the Introduction and Discussion which specific knowledge gap your work aims to fill and how it differs methodologically or conceptually from previous literature. In this regard, a more in-depth review of the current state of the art would strengthen the theoretical framework. Biological validation The manuscript would benefit from a discussion on the possible biological implications of the findings. It would be useful to consider how the observed mechanical stability might translate into graft behavior after implantation, in terms of integration, tissue remodeling, or resistance to complications. Additionally, the inclusion (or at least the future consideration) of histological or immunohistochemical analyses would add significant value to the overall tissue characterization. Study limitations Although some limitations are acknowledged, I would suggest briefly expanding the discussion on whether a six-month storage period is sufficient to assess potential mechanical degradation. In clinical practice, many grafts are stored for longer durations, and reflecting on this aspect would offer a more comprehensive view of the study’s scope. In summary, I believe this is a solid and promising piece of work. With the suggested improvements, the manuscript could become even more informative and valuable to the scientific and clinical community interested in vascular allografts and cryopreservation.  

Author Response

Please see the attachment for detailed answers and the updated manuscript with track changes.

Round 2

Reviewer 2 Report

Comments and Suggestions for Authors

In its present form, the manuscript represents a solid contribution to the mechanical characterisation of human allografts, with potential implications for logistical management in tissue banks.
While it does not provide functional data, it adequately fulfils its stated objective and has been substantially improved in this version.

However, I would like to suggest one final improvement before the manuscript is considered for publication:

I recommend adding a clear statement in the Discussion section emphasizing that, while the mechanical findings are valuable, they do not permit direct conclusions regarding the biological viability or functional integration of the graft following implantation. This clarification would help prevent potential overinterpretation by readers and reinforce the scientific transparency of the study. Additionally, I encourage you to enhance the depth of the discussion, particularly by further contextualizing the implications and limitations of your results.

Author Response

Dear Reviewer,

Please see our response to your comments in the attached letter and the updated manuscript with tracked changes. 
